



# Stationary and Portable Multipollutant Monitors for High Spatiotemporal Resolution Air Quality Studies including Online Calibration

Colby Buehler[1,2], Fulizi Xiong[1,2], Misti Levy Zamora[2,3], Kate M. Skog[1], Joseph Kohrman-Glaser[4], Stefan Colton[4], Michael

McNamara[5], Kevin Ryan[5], Carrie Redlich[6,7], Matthew Bartos[8], Brandon Wong[8], Branko Kerkez[8], Kirsten Koehler[2,3], Drew R.

Gentner[1,2,9,*]

[1] Department of Chemical & Environmental Engineering, Yale University, School of Engineering and Applied Science, New Haven, Connecticut 06511, USA

[2] SEARCH (Solutions for Energy, Air, Climate and Health) Center, Yale University, New Haven, CT, USA

[3] Department of Environmental Health and Engineering, Johns Hopkins Bloomberg School of Public Health, Baltimore, MD, 21205, USA

[4] Department of Mechanical Engineering, Yale University, School of Engineering and Applied Science, New Haven, Connecticut, 06511, USA

[5] Department of Electrical Engineering, Yale University, School of Engineering and Applied Science, New Haven,
Connecticut, 06511, USA

[6] Department of Internal Medicine, Yale University, School of Medicine, New Haven, Connecticut, 06510, USA

[7] Department of Environmental Health Sciences, Yale University, School of Public Health, New Haven, Connecticut, 06511, USA

[8] Civil and Environmental Engineering, University of Michigan, 2350 Hayward St, G.G. Brown Building, Ann Arbor, MI
48109, USA

[9] Max Planck Institute for Chemistry, Mainz 55128, Germany.

*Correspondence to*: Drew R. Gentner (drew.gentner@yale.edu)

**Abstract.** The distribution and dynamics of atmospheric pollutants are spatiotemporally heterogeneous due to variability in
emissions, transport, chemistry, and deposition. To understand these processes at high spatiotemporal resolution and their implications for air quality and personal exposure, we present custom, low-cost air quality monitors that measure concentrations of contaminants relevant to human health and climate, including gases (e.g. $O_3$, $NO$, $NO_2$, $CO$, $CO_2$, $CH_4$, and $SO_2$) and size-resolved (0.3–10 µm) particulate matter. The devices transmit sensor data and location via cellular communications, and are capable of providing concentration data down to second-level temporal resolution. We produce two
models; one designed for stationary (or mobile platform) operation, and a wearable, portable model for directly measuring personal exposure in the breathing zone. To address persistent problems with sensor drift and environmental sensitivities (e.g. relative humidity and temperature), we present the first online calibration system designed specifically for low-cost air quality sensors to calibrate zero and span concentrations at hourly to weekly intervals. Monitors are tested and validated in a number of environments across multiple outdoor and indoor sites in New Haven, CT, Baltimore, MD, and New York City. The



evaluated pollutants ($O_3$, $NO_2$, NO, CO, $CO_2$, and $PM_{2.5}$) performed well against reference instrumentation (e.g. r=0.66–0.98) in urban field evaluations with fast e-folding response times ($\leq$ 1 min), making them suitable for both large-scale network deployments and smaller-scale targeted experiments at a wide range of temporal resolutions. We also provide a discussion of best practices on monitor design, construction, systematic testing, and deployment.

## 1 Introduction

Exposures to air pollution are associated with elevated health risks such as cardiorespiratory inflammatory responses and oxidative stress (Brauer et al., 2012; Chuang et al., 2007; Pope and Dockery, 2006). Each year outdoor air pollution leads to approximately 4.2 million premature deaths worldwide and is the fifth highest mortality risk factor in the Global Burden of Disease Study 2015 (Cohen et al., 2017; Forouzanfar et al., 2016). Assessment of public health risks and regulatory standards requires accurate measurement of air pollution levels. However, traditional analytical techniques for air pollutant

measurements—such as spectroscopy, chemiluminescence, and mass spectrometry—are expensive, which limits the deployment of instruments to sparsely-located state and federal air quality monitoring sites and targeted research campaigns. As a result, the spatiotemporal variations in urban human exposure caused by localized combustion sources (e.g. motor vehicles, cooking) and other sources are not well understood (Kheirbek et al., 2013).

The need for better geospatial coverage in air quality monitoring has resulted in multiple studies that utilize low-cost sensors

to measure a range of pollutants in portable and stationary configurations (Bigi et al., 2018; Castell et al., 2017; Cross et al., 2017; Hagan et al., 2018; Lewis et al., 2016; Mead et al., 2013; Popoola et al., 2016; Thorson et al., 2019; Zimmerman et al., 2018). Wearable devices containing sensors to measure real-time gas-phase air pollutants such as nitric oxide (NO), nitrogen dioxide ($NO_2$), carbon dioxide (CO), carbon dioxide ($CO_2$) and ozone ($O_3$) have been developed, but proper calibration still poses a challenge (Cao and Thompson, 2016; Mead et al., 2013). For example, Piedrahita et al. (2014) developed wearable air

quality monitors (M-Pods) from primarily metal oxide semiconductor sensors. They demonstrated an ability to quantify ambient concentrations of CO, $NO_2$, $CO_2$, and $O_3$, and found that field calibrations, given a wider range of environmental conditions, performed better than laboratory calibrations.

On a larger scale, environmental compliance and policymaking require an understanding of the air pollutant sources and their transport processes, and long-term high-density stationary monitoring networks are needed to fulfill this purpose. In 2010,

Mead et al., (2013) deployed 46 sensor nodes in the Cambridge (UK) area for 2.5 months to measure NO, $NO_2$ and CO. This study demonstrated the feasibility of using low-cost sensors to obtain environmental data at high spatiotemporal resolution. A more recent deployment of two Aerodyne ARISense systems collocated with state air quality stations was described by Cross et al., (2017). This study reported mixed performance of Alphasense electrochemical NO, CO, $NO_2$ and $O_x$ ($r^2$=0.88, 0.84, 0.69, and 0.39) sensors at a 5 min temporal resolution. Numerous state, federal, and international programs continue to evaluate

emerging sensor technologies (e.g. South Coast Air Quality Sensor Performance Evaluation Center (AQ-SPEC), EPA Air





Sensor Toolbox, and the World Meteorological Organization (WMO)). While low-cost sensors have great potential to provide air quality data at higher spatiotemporal resolution and complement existing monitoring sites, multiple studies have reported measurement biases caused by sensor drift due to environmental variables and aging (Borrego et al., 2016; Cross et al., 2017; Lewis et al., 2016; Mead et al., 2013; Zimmerman et al., 2018). Hence, careful sensor characterization, calibration and data

processing are important to ensure measurement accuracy.

In this study, we design and evaluate custom stationary and portable multipollutant monitors as part of the SEARCH (Solutions for Energy, AiR, Climate and Health) Center at Yale-Johns Hopkins, which will deploy the multipollutant monitors in a long-term, city-wide network in Baltimore, MD. Specifically we: (a) describe the physical hardware design, sensors employed, and relevant testing with a focus on overcoming historical limitations; (b) describe and demonstrate the first online calibration

system for low-cost monitors; (c) present laboratory tests and field measurements with our monitors to demonstrate real-world performance; (d) compare the performance of the multipollutant monitor to other low-cost monitor deployments; and (e) provide best practices for monitor design to enable future research.

## 2. Materials & Methods

### 2.1 Instrument Design

#### 2.1.1 Incorporated Sensors

A suite of sensors is built into the SEARCH multipollutant stationary monitors to measure the concentration of CO, $NO_2$, NO, $CO_2$, $O_3$, methane ($CH_4$), size-resolved particulate matter (PM), and, when applicable, sulfur dioxide ($SO_2$). It also collects relative humidity (RH) and temperature (T) data to correct for RH/T influences on sensor response during field deployment with changing environmental conditions. Manufacturers and part numbers for the selected sensors are listed in Table 1. Due

to size limitations, either a NO or $SO_2$ sensor is included in the multipollutant monitor based on the application. The monitors evaluated in this study contain NO sensors to better characterize urban $NO_x$ (NO and $NO_2$) pollution. Monitors with $SO_2$ sensors will be applied specifically in targeted emissions studies or future locations with higher $SO_2$ concentrations. The $CH_4$ and $CO_2$ sensors are integrated to evaluate greenhouse gas emissions. The portable monitor integrates sensors for CO, $CO_2$, $NO_2$, $O_3$, and PM to evaluate personal exposure in a space-efficient package.

Measurements of CO, $NO_2$, NO, $O_x$, and $SO_2$ are made using the 4-electrode electrochemical ppb-level A4 series sensors from Alphasense (http://www.alphasense.com). Different models of electrochemical sensors manufactured by Alphasense have been tested in previous studies (e.g. Hagan et al. (2018), Mead et al. (2013), Zimmerman et al. (2018)) and demonstrate promise for ambient measurements with careful calibration and system design. For $O_x$, we use the updated B431 model. The 4-electrode configurations were chosen over the 3-electrode sensors because the extra auxiliary electrode (AE), which has the same





functionality as the working electrode (WE) but is not exposed to the analyte, provides a background electrode response. When coupled together with the WE, this reduces the influences of RH/T on sensor signals.

The Alphasense NDIR sensor measures $CO_2$, having an estimated limit of detection (LOD) of 1 ppm (Hodgkinson et al., 2013). The NDIR sensor has a broadband light source, and two bandpass filters centered at 4.26 µm and 3.95 µm. The 4.26 µm filter coincides with the $CO_2$ absorption band centered at 4.2 µm. The 3.95 µm light is not absorbed by $CO_2$ and works as a reference

to account for potential drift in light intensity caused by lamp aging and power supply change. The $CO_2$ sensor has similar dimensions as the A4 electrochemical sensors.

$CH_4$ measurements are made using the Figaro TGS2600 gas sensor. Field evaluations of this sensor were performed by Eugster and Kling (2012) in Alaska. More recently, van den Bossche et al. (2017) conducted a systematic laboratory evaluation of a similar Figaro model. Both groups reported measurement agreement between the sensor and a reference technique after

correcting for RH and temperature interferences. The Figaro TGS2600 is also sensitive to analytes such as CO, hydrogen ($H_2$), and volatile organic compounds (VOCs), such as ethanol and isobutane (manufacturer's specification). The cross sensitivity from CO can be corrected in the multipollutant monitor by using the onboard CO sensor. We remove the VOC interference by adding a layer of activated charcoal-impregnated cloth on top of the sensor to filter VOCs. The implementation and performance of this setup is detailed in Sect. 2.1.3.

For the stationary multipollutant monitor, the MiCS-2614 sensor measures $O_3$ due to its proven past performance, low cost, and small size (5 mm x 7 mm x 1.55 mm). This sensor was built into a portable ozone monitor by Cao and Thompson (2016) where they found it agreed with 2B Technologies' ozone monitor in the range of 20 ppb to 100 ppb, with over-measurement under 20 ppb and under-measurement above 100 ppb. Note: at the time of this publication, this sensor is not being manufactured.

PM is measured with a miniature PM sensor PMS A003 produced by Plantower (http://www.plantower.com). The sensor has an internal laser and uses scattered light to count particles and differentiate particle size. The device reports mass concentrations in $PM_1$, $PM_{2.5}$, and $PM_{10}$ with precision of 1 µg/m$^3$, as well as particle number concentrations for particle sizes bins: 0.3 µm, 0.5 µm, 1 µm, 2.5 µm, 5 µm, and 10 µm. Levy-Zamora et al. (2019) demonstrated the ability of the sensor to perform under laboratory and ambient settings. With environmental correction factors, the sensor had an overall accuracy of 93% and an

overall precision error of 10%.

### 2.1.2 Electrical System

The electronics for the multipollutant monitor are designed to have modularized functions on each individual circuit board. Each sensor has its designated analog circuitry to supply power, amplify signals, and filter noise. The analog signals are fed to analog-to-digital converters (ADC) on the daughter boards to minimize noise pick-up in the wiring or other circuitry.





The Alphasense electrochemical sensors are powered with potentiostatic circuitries with zero bias for the CO, $NO_2$, and $SO_2$ sensors, and a 200 mV bias for the NO sensor. Special care was taken to match the input impedance for the NO potentiostatic circuit to minimize noise. The circuit amplification is designed to output an analog signal of approximately 1 volt per 100 ppb NO, $SO_2$, $NO_2$, and 10 ppm CO. The onboard ADC sequentially converts the amplified and filtered signals generated by the AE and the WE. The AE voltage is recorded as the background signal, and the differential signal between WE and AE

voltages is used as the sensor signal for calibration and measurement purposes.

The $CO_2$ sensor is driven with a 2 Hz 5V 50% duty cycle waveform clocked by a MEMS (Micro-Electro-Mechanical Systems) oscillator. The outputs of the $CO_2$ sensor are two DC-biased sinusoidal waves from the reference and active channels, and subsequent circuitries are implemented to remove the DC offset and amplify the signals. Two peak detection circuits are applied to sample and hold the peak heights of the two amplified sinusoidal waves to be read sequentially by the ADC. This

design uses significantly less processing resources, in comparison with continuous sampling and peak detection through software.

The $CH_4$ and $O_3$ circuitries are placed on one circuit board to conserve space and accommodate mechanical requirements (Sect. 2.1.3). These two sensors function by changing their resistances when exposed to their corresponding analytes. Hence, voltage dividers with low temperature coefficient load resistors were applied, and the sensor resistances can be derived by sampling

the voltages across the load resistors through ADCs.

The humidity/temperature (RH/T) sensor is placed on a separate small circuit board and towards the front of the inlet to minimize the influence of heat generated by the voltage drop across circuit board traces in the presence of other components. The PM sensor is equipped with a circuit board to convert from its 1 mm pitch connection to a more convenient 2.54 mm pitch connection to facilitate assembly. The RH/T sensor and the PM sensor both output digital signals, and the signals are acquired

by the microcontroller directly. Daughter boards for the portable multipollutant monitor are combined or miniaturized versions of the stationary design in order to reduce the amount of wiring and required space.

A central control board generates sensor input voltages, powers components on/off (such as solenoid valves to perform calibration and background measurement), powers a piezoelectric blower (to circulate ambient air for the gas sensors), and reads, processes, stores and transmits sensor data. The control processes use the Cypress 68-pin PSoC 5lp microcontroller,

which interfaces with sensors through digital communication peripherals (I2C and UART). The data acquisition frequencies are set as following: the $NO_2$, NO, $SO_2$, and CO sensors are sampled every 160 ms, with AE and WE signals each taking up 80 ms sequentially; the $CH_4$ and $O_3$ sensors are sampled every 160 ms, utilizing only one signal channel; the RH/T sensor is sampled every 160 ms for either RH data and temperature data sequentially, making their actual sampling period 320 ms; the $CO_2$ sensor is sampled with 2 Hz frequency in accordance with the input drive frequency for both the active and reference



channels; and the PM sensor is sampled every 640 ms, to accommodate its low data output rate relative to the other components.
See Table S1 for more information regarding the electronic system.

### 2.1.3 Mechanical Design

The sampling manifold is designed to isolate the sensing areas of the gas sensors in a small active flow area separated from
the rest of the device components (Fig. 1, S3). The manifold is 3D-printed with WaterShed XC11222 resin through
stereolithography (SLA), which prints materials with a dense, gas-tight finish. Other 3D printing materials such as acrylonitrile
butadiene styrene (ABS) and polylactic acid (PLA) were also tested. These materials are often printed with the fused deposition
modeling (FDM) method, creating porous parts that need surface treatment of acetone to be gas-tight. To minimize the potential
shape deformation resulting from post-printing treatment, we use the SLA method to print the manifold and other 3D printed
parts of the device. O-ring grooves are incorporated in the manifold to secure and provide an air-tight seal for the sensors. To
minimize potential ozone loss, the ozone sensor is placed closest to the manifold inlet. In our testing with a 2B-Tech reference
ozone monitor, the ozone loss rate is 4–12% for XC11122 resin, versus 7–22% for ABS. To further reduce losses of reactive
analyte, a PTFE liner is inserted into the inlet of the XC11122 manifold to reduce contact between the sampling air and the
manifold material. The outlet of the manifold is connected to the piezoelectric blower that sampled at an average flow rate of
0.6 standard liters per minute (SLPM). To optimize monitor response time, the pollutant exchange rate in the manifold is
maximized with high flow rates and a small internal volume of ~9 ml, producing an estimated 1 s residence time in the
manifold. The ambient air entering the manifold is first pulled through a filter holder with a 2 µm thick, 47 mm diameter Teflon
filter to keep the inside of the manifold clean of particles and collect filter samples for offline analysis.

The $CH_4$ sensor inside the manifold is covered by a layer of activated carbon-impregnated cloth (Zorflex® Double Weave),
which is secured by a 3D-printed PLA cylindrical shell. It is then wrapped in Teflon tape in order reduce pollutant interactions
with PLA inside of the manifold. This charcoal cloth layer is effective in filtering out VOC interference for the sensor. For
instance, when covered by the charcoal cloth, the $CH_4$ sensor did not respond to ethanol concentrations as high as 2%. Even
after continuous exposure to outdoor VOC for 3 months, with the charcoal cloth cover, the $CH_4$ sensor did not respond to
ethanol vapor when an open vial was placed near it. For comparison, when the $CH_4$ sensor with the used charcoal cloth was
placed directly above an open vial of ethanol, sensor resistance dropped by approximately 5 kΩ, equivalent to 0.3 ppm
methane. While such highly concentrated ethanol vapors are less common in the ambient environment, the activated carbon
filter is also likely effective for other VOCs at lower concentrations.  Here, ethanol is specifically tested as the challenge
compound because the sensor is known to be highly responsive to ethanol and activated carbon is a known, effective
hydrocarbon filter for a wide range of VOCs.

Inlet and outlet enclosures are designed for the PM sensor to direct air flow (Fig.1). Specifically, the inlet enclosure contained
an SLA 3D-printed Watershed XC11122 holder to support the sensor and an aluminum inlet, through which sample air flows



into the sensor inlet. Aluminum was chosen over 3D-printed plastic material as the inlet duct and grounded to the motherboard to avoid electrostatic particle losses due to static charges on a non-metallic surface. The front of the aluminum duct is covered with an aluminum disk placed 30 mm above it, between which a 32 x 32 mesh stainless steel wire cloth is installed to block insects and large dust particles. Ambient air flows through the screen and enters the aluminum channel to reach the sensor

inlet. The aluminum disk is placed above the inlet to block light, which was shown to interfere with normal operation and cause the sensor to output PM mass concentrations above 3000 µg/m$^3$. To reduce the intrusion of light and water, the device is installed with both the gas and particle inlets pointed downwards.

The portable monitor makes all measurements immediately adjacent to the breathing zone with all sensors contained in a small custom shoulder-mountable housing that is 3-D printed and easily attachable to a bag, backpack, purse, or other strap (Fig. 1c–

d). A small auxiliary enclosure (23 cm x 12 cm x 6.5 cm; 1 kg including battery) is required to house the re-chargeable battery and main circuit board. The design of the gas sensor manifold is similar to that of the stationary monitor with a piezoelectric blower promoting fast air exchange rates in a minimal volume manifold with PM removed at the inlet via a 23 mm PFTE filter in a PTFE housing. PM is measured via a separate minimal inlet with a light shield.

### 2.1.4 Online Calibration and Zero System

All monitors undergo multi-point calibrations under variable, realistic RH/T conditions in an environmental chamber prior to field installation. To improve data quality during field deployment and to better track and correct for sensor drift, the stationary monitor includes a laboratory tested span calibration and zero system. These systems are not incorporated into the portable model to conserve space and minimize weight. However, portable monitors will be periodically calibrated in the environmental chamber over the course of the SEARCH project.

The calibration process has either two or three calibration functions depending on the configuration, with the ability to change the temporal frequency at which each occurs. Each stationary monitor has zeroing functions for both the PM sensor and the gas sensor suite while half of the monitors for the SEARCH deployment also have a gas span calibration function using a miniaturized standard cylinder (dependent upon cylinder composition). In the SEARCH project, the PM and gas zeroing functions are scheduled to occur twice a week and the gas span calibration once a week. Depending on the inclusion of the

standard gas cylinder, two or three 3-way solenoid valves are placed in the system to direct flows and alternate normal ambient sampling with PM zero, gas zero, and gas span calibration functions (flow diagram shown in Fig. S4).

For the PM zero, the exhaust from the piezoelectric blower of the gas system, in which particles have been filtered out by a Teflon filter, is directed to the aluminum inlet of the PM sensor. This results in an overflow to the PM inlet due to the higher flow rate of the gas system relative to the particle system, producing a mass concentration of zero.





Zero concentrations for the gas sensors are obtained using either a filtering "zero trap" or via their absence from the calibration cylinder, which is primarily nitrogen and specifically useful for compounds that do not have room-temperature filtration options available for the zero trap. A series of scrubbing materials are used to remove select gas-phase analytes: soda lime for $CO_2$ and activated carbon and stainless-steel wool for $O_3$. To obtain the zero-concentration signals with the zero trap, the exhaust of the piezoelectric blower is passed through the packed materials directed to the gas sensors through a side port on

the manifold near the inlet. The flow rate through the packed tube is 50 standard cubic centimeters per minute (sccm). With the 9 ml internal volume, the air inside the manifold is re-circulated and passed through the packed tube 16 times in 3 min to ensure complete analyte removal. At the time of writing, we were unable to find materials to effectively remove $CH_4$, CO, $NO_2$, and NO at ambient temperatures. Therefore, their zero-concentration signals are determined in the laboratory with zero air prior to field deployment and are checked routinely with the balance of zero air in the calibration cylinder (with the

exception of CO and $CH_4$, which are calibrated via the CO present in the cylinder).

The standard gas delivery system is designed to overflow the manifold with known concentrations of gas standard from a miniature stainless steel gas cylinder (2'' OD x 5.5", Swagelok) that is filled with 5 ppm CO and 2000 ppm $CO_2$ to 1500 psig in a balance of nitrogen from a primary authentic cylinder (Airgas). Prior to installation, the pressure regulator (Tescom) is adjusted, in combination with a 0.006"–0.007" ID PEEK constriction (1/16" OD), to deliver >30 sccm of standard gas flow

into the manifold through the exhaust port of the piezoelectric blower with the blower off. Delivery at or above that flowrate is used to ensure constant overflow conditions regardless of cylinder pressure.

A water permeation setup is included in the standard gas delivery line to maintain a minimum humidity inside the manifold during calibration and prevent unrealistically dry conditions that skew electrochemical sensor response. The water permeation device includes a 3/8" diameter PTFE membrane (0.001" thick, McMaster-Carr) installed in a stainless steel tee (Swagelok)

where the membrane separates the standard gas flow from a reservoir of deionized water. The PTFE membrane restricts direct water flow but allows for sufficient permeation of $H_2O$ molecules to raise the humidity of the dry standard gas to a level (>40% in this study) that allows for acceptable calibration conditions. Furthermore, with field operation, RH conditions during calibration can be routinely monitored using the RH/T sensor reporting via the data network.

### 2.1.5 Cellular Communications and Data Storage

All raw sensor data is written to a local SD card at sub-second frequency. Every 10 s the data is averaged and transmitted to a database hosted on a cloud server through an onboard 4G Telit LE910C1-NS cellular module. To achieve fast and continuous sensor data collection while maintaining simultaneous cellular data transmission, a task preemptive scheduler implemented within the microcontroller firmware tracks the status of the sensors and cellular module and executes core processes (e.g. read, write, send, and receive) at pre-set time intervals.  This ensures that all sensor measurements are prioritized over other operating

system tasks.



The data streams stored on the SD card and the cloud server include differential, working, and/or auxiliary channels for electrochemical and NDIR sensors, resistances for metal oxide sensors, size-resolved PM mass and number concentration, power supply voltage, and diagnostic information for calibration processes. The InfluxDB time-series database (www.influxdata.com) is used to store, receive and serve sensor data from field-deployed monitors. To maximize data availability, the database is hosted on an Amazon Web Services Elastic Compute Cloud (AWS EC2) instance (aws.amazon.com). The open-source data visualization platform Grafana (grafana.com) allows users and stakeholders to see real-time field-deployed monitor data remotely through a web browser (see Fig. S5 for screenshots of these data platforms).

## 2.2 Instrument Evaluation

Our multipollutant monitor undergoes two phases of evaluation in this study: laboratory chamber experiments and ambient co-location with reference instruments. The laboratory experiments provide an isolated environment for characterizing sensor performance and establishing signal-response-to-concentration calibration curves. Outdoor ambient co-location experiments with reference instrumentation test the performance of the monitors over extended periods of time under a variety of real environmental conditions.

For laboratory calibration experiments at Johns Hopkins University, multipollutant monitors are placed inside a custom-built steel chamber (0.71 m x 1.35 m x 0.89 m). Environmental conditions range from 5–85% humidity (most occurring around 30–50%) and 20–40°C. Each gas pollutant is introduced into the chamber through filtered air inlets and diluted to a variety of concentrations above and below typical urban ambient levels using zero air (see Table 2). $PM_1$, $PM_{2.5}$, and $PM_{10}$ are evaluated using the methodology presented in Levy-Zamora et al. (2019). Online calibration system tests and other sensor response tests at Yale University involve supplying authentic gas standards (Airgas) to the multipollutant monitor inlet or a similar sensor housing. See the SI for linear calibration data (Fig. S6–7) examples and information on how concentration values are calculated for electrochemical sensors.

Field evaluation took place at 3 different locations, dependent on the availability of reference instruments for inter-comparison. They included near an arterial roadway on Yale's campus in New Haven, CT (Wall St.; 3/26/2018–4/7/2018); Baltimore, MD at the State of Maryland Department of the Environment Oldtown site (Oldtown Fire Station, 1100 Hillen St.; 5/18/2017–6/7/2017, 11/2017–12/2017, and 6/14/2018–7/12/2018); as well as in New York, NY (6/23/2018) and Baltimore, MD (3/2/2019) for separate tests of the portable monitor.

The temporal resolution of the comparison between the multipollutant monitor and reference instrumentation is primarily limited by the reference instrument. For $NO_2$, $PM_{2.5}$, and $CO_2$ evaluations made at the Oldtown site in MD, the lowest temporal resolution of the reference is 1 hr. For the New Haven, CT field evaluations we use an on-site 2B-Tech (model 202) for $O_3$ and Thermo Scientific Model 48i for CO and are able to show comparisons at finer time resolution; 1 min for $O_3$ and 10 min for CO. For comparison to the other sensors and literature, $O_3$ and CO are also reanalyzed at a 1 h average. In New Haven, 1





h NO reference data is used from the Connecticut Department of Energy and Environmental Protection (DEEP) Criscuolo Park site.

Common metrics for evaluating the performance of sensor configurations include linear regression parameters between RH/T
corrected sensor data and reference instrument data such as the coefficient of correlation (r), coefficient of determination ($r^2$), slope (m), and intercept (b). Ideal sensor performance would show strong correlation (r=1 or -1, $r^2$=1) as well as minimal over-or-under estimation of the true concentration (m=1, b=0). Statistical error tests such as the mean bias error (MBE), mean absolute error (MAE), and root mean square error (RMSE) are also commonly used. The MBE represents the tendency for the sensor to over- or under-estimate the reference, although positive and negative errors can cancel each other out. To get around
that limitation, the MAE is similar to the MBE but looks only at the average absolute difference between the sensors. Finally, the RMSE represents how narrow the error distribution is by penalizing large measurements errors. All tests are reported in concentration units which allows for physical interpretation of sensor performance. They are calculated as following:

$$MBE = \frac{1}{n}\sum_{i=1}^{n}\left(C_i^{MPM} - C_i^{ref}\right), \tag{1}$$

$$MAE = \frac{1}{n}\sum_{i=1}^{n}\left|C_i^{MPM} - C_i^{ref}\right|, \tag{2}$$

$$RMSE = \sqrt{\frac{1}{n}\sum_{i=1}^{n}\left(C_i^{MPM} - C_i^{ref}\right)^2}, \tag{3}$$

where $n$ is the total number of co-location data points, $C^{MPM}$ is the concentration value of the multipollutant monitor, and $C^{ref}$ is the concentration value of the reference monitor.

## 3 Results and Discussion

### 3.1 Stationary Monitor Field Results

**3.1.1 Particulate Matter (PM2.5):** Comparison of the Plantower PM sensor at the Oldtown site (Baltimore, MD) shows strong correlation (r=0.91, m=1.0) for $PM_{2.5}$ with a 1 h averaging window over the span of 4 weeks. Figure 2a shows a clear overestimate in raw $PM_{2.5}$ by the Plantower sensor at higher concentrations as reported in Levy-Zamora et al. (2019), where we derive a laboratory RH/T-correction equation to reduce bias and error (in this study: MBE=+0.9 μg/m³, RMSE=4.3 μg/m³). At a separate New Haven, CT deployment, our $PM_{2.5}$ measurements are well-correlated (r=0.94–0.98, m=1.01–1.33) between
five different multipollutant monitors at a 10 min resolution (Fig. 2b). See Fig. S8 for the time series data from Fig. 2b.

**3.1.2 Nitrogen Dioxide:** The $NO_2$ sensor exhibited strong correlation (r=0.88, m=0.93) at the Oldtown site at hourly resolution. The monitor tracked with the reference well during both clean periods and periods of pollution maxima (MBE=+0.8 ppb, RMSE=5.3 ppb), with concentrations ranging from near zero to over 50 ppb during the deployment (Fig. 3a). Correlation





values between the raw multipollutant monitor data and the reference are significantly improved using a RH/T correction (Fig.

3b; see SI for RH/T-correction procedures). In addition, 35% of data points fall within 10% of the reference instrument and 70% fall within 30% (Fig. 3c). The $NO_2$ sensor is known to be cross sensitive to $O_3$, but the $NO_2$ sensors are manufacturer-equipped with an $O_3$ filter rated to withstand 500 hours at 2 ppm (or longer at lower concentrations). While the $NO_2$ sensor did not exhibit cross sensitivity during the Oldtown deployment due to low ozone concentrations, future deployments should take the rating of the ozone filter into consideration and routinely monitor for biases due to ozone.

**3.1.3 Carbon Monoxide:** The CO sensor demonstrated strong correlation (r=0.92, m=1.2) in a week-long deployment in New Haven, CT at 10 min resolution. Figure 4a shows good tracking of pollution events and background shifts where concentrations exceeded 400 ppb but a noticeable underreporting relative to the reference during these relatively clean periods (<200 ppb). These deviations coincided with elevated temperatures inside the monitor (>15°C) which is consistent with the zero temperature dependence listed by the manufacturer (Alphasense CO-A4 Data Sheet). To correct some of this offset, two sets

of temperature corrections were used: a linear fit for all readings above 18°C and a linear fit for all readings below 18°C. Logarithmic and quadratic fits were tested for the high temperature relationship, but the best fit was linear. See Fig. S9 for the low and high temperature data points. Figure 4b shows that even with the correction factors, the overall trend and error (MBE=+5 ppb, RMSE=59 ppb) remain similar to the raw data. After these corrections 43% of data points were within 10% of the reference and 88% were within 30% at a 10 min resolution (Fig. 4c).

**3.1.4 Carbon Dioxide:** $CO_2$ showed moderate correlation (r=0.66, m=0.59) with the NIST North-East Corridor Project's NEB tower site over a three-week deployment at a 1 h resolution. While not a direct co-location (2.7 km apart), the NEB site is used as a reference to examine city-wide $CO_2$ levels and trends while acknowledging that spatial differences from local sources may limit the inter-comparison due to vertical or horizontal variance. Figure 5a shows the monitor trends well with the reference after temperature-correction, although the monitor occasionally exceeds the reference concentration by 10-20 ppm

(MBE=+3.4 ppm, RMSE=11 ppm). This is consistent with the reported accuracy of ~15 ppm shown in the manufacturer data sheet at 400 ppm in laboratory testing (Alphasense IRC-A1 Data Sheet). Possible lags in regional pollution episodes can be observed in the time series data (e.g. 6/30, 7/1, 7/7) leading to a lower correlation value than other presented pollutants. Despite this, 70% of readings were within 2.5% of the reference instrument (~10 ppm) and 98% were within 7.5% (~30 ppm).

**3.1.5 Ozone:** The $O_3$ sensor exhibited strong correlation (r=0.97, m=0.99) at a high temporal resolution of 1 min in New

Haven, CT. The raw sensor resistance followed the ozone concentration measured by the 2B monitor with an exponential relationship (Fig. 6b). With this fitted exponential curve, the ozone concentration from sensor measurement was derived, and compared to the 2B monitor results to evaluate the sensor's performance and dependency on environmental factors (MBE=-0.2 ppb, RMSE=3.3 ppb). We found that at low ozone concentrations (<10 ppb), there is considerable measurement discrepancy between the two devices (Fig. 6c-d). This is consistent with the sensor manufacturer's 10–1000 ppb rating for the

device's measurement range (Table 1). For ozone concentrations higher than 10 ppb, 67% of the data points agree within 10%,





and 99% within 30% of the reference (Fig. 6e). It is worth noting that the sensor output was not significantly affected by changes in RH during the study, despite large variation in these environmental conditions (Fig. 6d). The temperature effect is more significant with the concentration ratio for our monitor to the reference (for data >10 ppb), following the relationship *0.85 + 0.70 exp(-0.11T)*. Still, the sensor's temperature biases are minor here as Fig. 6e presents non-temperature-corrected

results.

**3.1.6 Nitric Oxide:** Without an immediately co-located reference NO monitor, we compared the NO sensor performance against the near-road DEEP Criscuolo Park site (1.6 km away from sampling location in downtown New Haven). The sensor had higher error terms than other low ppb-level sensors (MBE=+1.6, RMSE=16 ppb) likely due to the distance difference and deviations due to local dynamics, but still showed good agreement (Fig. 7) and reasonable correlation (r=0.74, m=0.86; Fig.

S11) at a 1 h resolution. We also leverage $O_3$ and CO measurements during the same sampling window to understand NO concentrations as NO readily reacts with $O_3$ and is often co-emitted with CO during combustion. The two-week campaign shows the effect of high NO concentrations on $O_3$ abundances due to the reaction $NO + O_3$, and CO enhancements coincide with elevated NO concentration levels (Fig. 7). NO concentrations at 1 h resolution during the New Haven deployment range from 0 to 160 ppb with three major concentration enhancements. Each buildup occurred overnight and dissipated around mid-

morning, potentially owing to periods of decreased ventilation and the accumulation of NO from nighttime traffic emissions in the nocturnal boundary layer without any photochemistry. A 1 h time lag relative to the reference site is observed on some NO spikes which is also observed in the CO data, and is likely due to differences in sampling location. During the three periods when NO concentrations exceeded 100 ppb for an extended period of time, $O_3$ concentrations decreased to ~0 ppb and CO concentrations exceeded 600 ppb. It is worth noting that the NO sensor used does not have significant cross-response to $O_3$ or

CO (Alphasense NO-A4 Data Sheet).

**3.1.7 Others:** Other pollutants not discussed in detail here include $CH_4$ and $SO_2$. The Figaro TGS2600 methane sensor demonstrated high linearity in signal response in the laboratory (Fig. S7) and effective VOC filtration, but a reference monitor was not available and future testing is planned. While the multipollutant monitors field-tested in this paper do not include the $SO_2$ sensor, and it is not planned for the SEARCH deployment, our laboratory results (Fig. S6) and past work (Hagan et al.,

2018) suggest it is suitable for measurements in locations with $SO_2$ concentrations higher than typical urban levels (>15 ppb).

**3.2 Portable Monitor Field Testing**

Personal exposure data was collected using our shoulder-mounted portable monitor in Manhattan, New York City (Fig. 8), and Baltimore, MD (Fig. 9). These results are discussed in brief with a focus on $PM_{2.5}$. In New York City, $PM_{2.5}$ concentrations reached a maximum of 210 µg/m$^3$ at a restaurant where the average was 34 µg/m$^3$ while inside. Several closely

occurring spikes with $PM_{2.5}$ maxima at 30–175 µg/m$^3$ occurred in an area with food carts on Broadway Ave. between 34th and 57th St., including open flame meat cooking, small power generators, and cigarette smoking. An average concentration of 9 ± 21 µg/m$^3$ was encountered across the 6 h period that included a mix of indoor and outdoor environments, and a mix of parks





and streets. Concentrations while in parks were lower than when walking along streets. The average concentrations across the southern transect of Central Park (5:45 pm) and Madison Square Park (7:30 pm) were 0.5 µg/m$^3$ and 1.4 µg/m$^3$, respectively,

compared to 3.9 µg/m$^3$ while on the streets before and after. Coupling the high sampling rate of one measurement per second with RH/T correction factors, the portable monitor has the ability to capture high-temporal resolution events (i.e., 10–20 s) for $PM_{2.5}$. $PM_1$ had similar trends, with a $PM_{2.5}/PM_1$ ratio of 1.44 at the restaurant, 1.48 at the parks, and 1.44 for the entire study period. Due to the lack of a robust RH/T correction factor for $PM_1$, we report only the ratio for $PM_{2.5}/PM_1$ in the raw data.

A day-long deployment in Baltimore shows $PM_{2.5}$ concentrations vary widely across locations and transportation modes (Fig.

9) and can be mapped within a city via GPS. Elevated levels of 35 µg/m$^3$ occurred in the morning at a restaurant before reaching consistent values of less than 10 µg/m$^3$ for most of the day. There were occasional concentration spikes, such as walking through a commercial store (see purple symbols at 11:00am) where measurements rapidly rose from ~0 to 30 µg/m$^3$. In the afternoon, concentrations spiked to above 110 µg/m$^3$ while driving during rush hour with the windows closed. GPS functionality was maintained throughout the study and accurately depicted the path of the participant, yet some path

information in Fig. 9 is not depicted during vehicle transport due to averaging to 1 min intervals.

### 3.3 Sensor Response Time

A key performance characteristic of any field-deployed analytical instrument is its response time to changes in pollutant concentrations, especially in dynamic urban environments where concentrations change rapidly with source proximity or microenvironments. This response time is a function of air exchange rates within the sampling system and the individual sensor

response times, which are inherently limited in some sensors involving electrochemical processes. A useful metric to examine this is the e-folding time (i.e. a decrease to a signal of 1-1/e or ~63%) of sensor signals due to abrupt changes in pollutant concentration. A long e-folding time indicates a sluggish sensor, while a short e-folding time indicates a responsive sensor that can respond to a dynamic environment and distinguish changes at higher-temporal resolution. To characterize the sample delivery systems, Fig. 10 shows the response of several sensors in the multipollutant monitor and their e-folding times. $PM_{2.5}$

has the shortest e-folding time of roughly 10 s ($PM_1$ and $PM_{10}$ are similar) due to its optical sensing technique (in a separate sampling inlet). CO and $CO_2$ have similar e-folding times of 20 s, demonstrating an ability to capture changes at under 1 min resolution (see Fig. S10 for analysis of 20 s $PM_1$ and CO roadside plumes). NO and $NO_2$ take longer to respond, 50 and 65 s respectively, but are still capable of capturing urban dynamics at a 5–10 min resolution.

### 3.4 Online Calibration Processes

**3.4.1 Standard Gas Cylinder Delivery:** A laboratory test of the gas cylinder delivery system is shown in Fig. 11a delivering known concentrations of CO and $CO_2$ in a balance of nitrogen for a span check as well as zero concentration signals for other sensors (e.g. $NO_2$, NO). With the small internal volume of the manifold flushed by 30 sccm of standard gas, it quickly reaches stabilized signals within the 3 min calibration periods, consistent with expectations based on Fig.10. Five repeated runs at a



delivery pressure of 35 psig (prior to the constriction tubing that substantially reduces pressure to near 1 atm) demonstrated
consistent behavior and is representative of initial performance in field tested units. Expected shifts in RH are observed (and
can be used in-field to evaluate RH sensitivity), but the permeation device maintains humidity at relevant conditions despite a
completely dry standard gas.

**3.4.2 Zero Trap:** Figure 11b shows the effective removal of $CO_2$ using the gas zeroing function, while stable concentrations
are observed for non-zeroed gases. Changes in NO concentrations (and less so for $NO_2$) occur due to the observed changes in
RH. During long-term field deployment, RH changes occurring both during the use of the calibration gas cylinder and the zero
trap are also useful indicators to check in-field RH-dependent changes in response factors and zero signals, respectively, when
occurring under stable concentrations. Similarly, differences in temperature between zeroing periods can be used to check
temperature-related variations for sensors with significant temperature response. Figure 11b does not include $O_3$ data as the
levels in the laboratory were too low for sensor response to move off the baseline. Experiments at higher concentrations
indicate that stainless steel is an effective scavenger of $O_3$ (see Fig. S12).

**3.4.3 PM Zero:** Field testing of the PM zero method was conducted near food cart vendors in New Haven, CT. Ambient
concentrations ranged between 3 and 48 µg/m$^3$ with rapid changes due to moderate wind levels and proximity to active sources.
After switching the valves, PM levels were effectively zeroed for two minutes and rose back immediately to higher
concentrations and saw large spikes after the zero ended (see Fig. S13). For monitors in very close proximity to highly-
concentrated plumes (e.g., >250 µg/m$^3$), longer zero periods or scheduling during low activity periods may be necessary to
fully flush the inlet and avoid/isolate bias from concentrated plumes.

**3.5 Comparison with Literature**

To contextualize the performance of our monitors, Table 3 shows a summary of co-location statistical data with several recent
literature field deployments. For a more extensive comparison, see Karagulian et al. (2019). The presented performance metrics
are specific to the region and conditions they were evaluated in, and differences in sampling locations, environmental
conditions, pollutant mixtures, and testing durations should be considered in future applications. For $NO_2$, our multipollutant
monitor had a higher $r^2$ (0.77) than other studies except for Bigi et al. (2018) ($r^2$=0.80), with our MAE and RMSE lower by
2.1 and 3 ppb, respectively. The NO sensor shows lower correlation than other studies such as Cross et al. (2017) in terms of
$r^2$ (their $r^2$ of 0.84 compared to our 0.54) and error terms (their RMSE of 4.52 ppb compared to our 16 ppb), though we are
comparing against a reference instrument located 1.6 km away with evident localized dynamics (see Sect. 3.1.6). For CO, the
multipollutant monitor performed similarly to other studies using Alphasense electrochemical sensors, such as Zimmerman et
al. (2018), where our $r^2$ and MAE were lower by 0.11 and 3 ppb, respectively. To our knowledge this work is the first low-
cost urban air sensor network implementation of the Alphasense IRC-A1 sensor. Zimmerman et al. (2018) measured $CO_2$ using
an SST CO2S-A, but no reference instrument was available for their co-location study. One study that did report $CO_2$ co-



location results was Spinelle et al. (2017), which showed higher $r^2$ (0.51–0.79) but similar slope values after using an artificial neural network for calibration. For $O_3$, the deployments used for comparison primarily utilized a form of the Alphasense Ox-B4 sensor (in tandem with $NO_2$ electrochemical sensors). Our MiCS sensor performed well compared to other co-locations in regards to $r^2$ and m values, with only Zimmerman et al. (2018) (using the same updated Ox-B431 which we employ in our portable monitors) reporting similar MBE. Ripoll et al. (2019) used both an Alphasense Ox-B431 and MiCS-2614 sensor and

found that both exhibited strong performance, similar to our results. For $PM_{2.5}$, we compare our sensors primarily with selected results from AQ-SPEC testing (i.e. five highest $r^2$ values) from Feenstra et al. (2019). $PM_{2.5}$ sensors saw a wide range of $r^2$ values being reported (0.73–0.95) with the highest coming from the PurpleAir PA-II which uses a Plantower sensor and the two next highest being Plantower A003 sensors from this study. To summarize the performance distribution of our pollutant measurements, Fig. S14 compares the values from Table 3 to the "best selection region" defined by Karagulian et al. (2019).

**3.6 Best Practices**

The field of low-cost air quality sensors is rapidly improving, and new generations of monitoring devices should strive to further improve accuracy and precision. Here we present a list of lessons learned, obstacles faced, and recommendations for future design, fabrication, and deployment. Careful consideration of electronic design and sensor selection can eliminate complications later in the process. We make the following recommendations: choose low noise components where applicable

to enhance precision and improve detection limits; transform signals from analog to digital near to the sensor to preserve signal; utilize ADCs of sufficiently high resolution for the application to achieve the resolution necessary for the pollutant measurement application; use electronic shielding on sensitive sensing or signal transmission components; monitor power delivery in real time and report back auxiliary and supply signals; and measure RH/T dependent channels where applicable (e.g. auxiliary electrode on Alphasense sensors).

Good design should take into account sensor-to-sensor performance and practice good quality assurance and quality control (QA/QC) of sensors, both prior to installation and for a trial period after installation. For some of the sensors used in the multipollutant monitor, there was more deviation from lot-to-lot than anticipated which requires careful laboratory calibration to correct. To minimize the impact of sensor variability on field measurements, characterize sensors before installation and deployment for supplemental quality control purposes. If possible, compare new sensors against typical response patterns

found in the deployment already in order to gauge whether a sensor or circuit board may be malfunctioning early on. With a large network, some amount of automation will be necessary to quickly determine malfunctioning monitors. Also, carefully consider the position within the manifold and implement measures to keep the sensors clean and away from interferences, such as upstream particle filtration for gas-phase sensors or positioning the NDIR $CO_2$ sensor last to reduce the influence of waste heat, respectively.



Active flow is critical and allows us to achieve high temporal resolution measurements with e-folding times below one minute for most pollutants. In urban settings and for personal exposure studies this provides additional data points to identify rapidly changing emissions and environmental conditions. When deploying the monitors: use a water-tight enclosure with inlets pointed downwards to avoid light and water intrusion; shade them from direct sunlight if possible at a given site to reduce temperature swings that exacerbate temperature-dependent calibration changes; implement mesh to the inlets to prevent insects

from entering; and be mindful of point source emissions nearby to reduce undesired bias. Additionally, check for flow balance consistently during construction and then periodically in the field (during deployment or servicing) to (i) confirm high active flow rates, (ii) verify that measurements are not being compromised with air from within the enclosure, and (iii) ensure that on-board calibrations perform properly. Additionally, in order to better assess issues remotely, choice of wireless connectivity should be evaluated. Cell connections are convenient and allow for the monitor to be placed nearly anywhere but can be spotty

at times and can be more expensive to operate than a WiFi-based approach. Protocols for remote access into the monitors to address problems and prompt a restart are desirable to minimize wasted travel time to sites.

## 4 Conclusions

Our multipollutant monitors advance the state of the field by monitoring 9+ gas and size-resolved PM pollutant data streams simultaneously in an optimized fast-response active flow system. The stationary monitor includes a novel on-board calibration

system and the portable shoulder-mountable monitor samples in the breathing zone. We implement low-noise electronic design, GPS tracking, and cellular communications to communicate ambient and calibration data in real-time—all to enable more accurate and precise cost-effective monitor networks for stationary or mobile platforms. The calibration system is flexible and can be adjusted for a variety of analytes of interest via tuning of the calibration gas or zero trap. Additionally, a greater range of RH/T points can be gained by increasing calibration frequency with strategic timing of calibration functions across

the day. Still, there is a need for continual improvement of laboratory and field calibration procedures, effects from RH/T, and comparisons to reference instrumentation through permanent and temporary co-location. With most pollutants achieving high correlation in urban field evaluations, these systems are ready for large-scale network deployments and smaller scale targeted measurements.

### Data Availability

Data will be made available by the authors upon request.

### Supporting Information

Figures S1–14 and Table S1 are available in the Supporting Information (PDF), available free of charge online.





**Author Contributions**

FX, MLZ, BK, KR, KMS, JKG, KK, and DRG conceptualized the multipollutant monitor mechanical design, electronics,
and/or online calibration system. CB, FX, MLZ, JKG, SC, MM, and DRG designed and constructed the multipollutant
monitors or tested their components. CB, FX, MM, and MLZ performed calibration testing. CB, FX, MLZ, CR, and DRG
performed co-location experiments and/or collected data. CB, FX, and MLZ performed data analysis. MB, BW, and BK helped
design and develop electronics, cellular communications, data management. KR also provided guidance on the electronic
design and best practices. CB, FX, and DRG prepared of the original manuscript. All authors have reviewed the paper.

**Competing Interests**

DRG has externally-funded projects on low-cost air quality monitoring technology (EPA, HKF Technology), where the
developed technology has been licensed by Yale to HKF Technology.

**Acknowledgements**

We thank HKF Technology, the U.S. Environmental Protection Agency, and The Heinz Endowments for funding the research.
C.B., D.R.G., K.K., B.K., and M.L.Z. acknowledge support from the Assistance Agreement No. RD835871 awarded by the
U.S. Environmental Protection Agency to Yale University. It has not been formally reviewed by EPA. The views expressed
in this document are solely those of the authors and do not necessarily reflect those of the Agency. EPA does not endorse any
products or commercial services mentioned in this publication. C.B. is supported by the National Science Foundation Graduate
Research Fellowship Program under Grant No. DGE1752134. Any opinions, findings, and conclusions or recommendations
expressed in this material are those of the author(s) and do not necessarily reflect the views of the National Science Foundation.
M.L.Z. was also supported by the National Institute of Environmental Health Sciences of the National Institutes of Health
under awards number 1K23ES029985-01 and K99ES029116. The content is solely the responsibility of the authors and does
not necessarily represent the official views of the National Institutes of Health. The authors thank the Maryland Department
of the Environment Air and Radiation Management Administration for allowing us to colocate our sensors with their
instruments. For their assistance, we thank the Yale Center for Engineering Innovation and Design (CEID); Jordan Peccia
(Yale); the following Yale undergraduates for their assistance: Kevin Truong, Hannah Nesser, Patrick Wilczynski, Genevieve
Fowler, Mieke Scherpbier, Jonathan Chang, Christian White, Marissa Huiling Foo, Reese Roberts, Bohan Lou, Christian
Mendez, Tori Hass-Mitchell, Jinny Van Doorn, Pulith Peiris; and Amir Bond and Ethan Weed (Peabody EVOlutions). We
thank Alphasense for their help with sensor integration and signal processing.




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



.

**Table 1: Technical specifications concerning the pollutant and environmental sensors used.**

| Sensor Target | Manufacturer and Part Number | Method | Observed LOD |
|---|---|---|---|
| CO | Alphasense CO-A4 | 4-electrode electrochemical | 20 ppb |
| NO | Alphasense NO-A4 | 4-electrode electrochemical | 3 ppb |
| $NO_2$ | Alphasense NO2-A43F | 4-electrode electrochemical | 1 ppb |
| $SO_2$ | Alphasense SO2-A4 | 4-electrode electrochemical | 15 ppb |
| $O_3$ | Alphasense OX-A431 | 4-electrode electrochemical | 1 ppb |
| $CO_2$ | Alphasense IRC-A1 | Infrared, pyroelectric | [a] |
| $CH_4$ | Figaro TGS 2600 | Metal oxide resistance | [a] |
| $O_3$ | MiCS-2614 | Metal oxide resistance | 10 ppb |
| $PM_1$, $PM_{2.5}$, $PM_{10}$ | Plantower A003 | Optical particle counter | 1 µg/m$^3$ |
| RH & T | Sensirion SHT25 | - | 0-100 %, 0-120 °C[b] |

[a]Below background concentrations. [b]From manufacturer data sheet.

**Table 2: Lab calibration procedure and environmental condition ranges for gases. Each non-zero gas concentration is maintained for 90 minutes.**

| Gas | Concentrations/Ranges |
|---|---|
| NO | 0, 5, 10, 30, 50 ppb |
| CO | 0, 1, 2, 3, 4 ppm |
| $O_3$ | 10, 20, 50, 75, 100 ppb* |
| $NO_2$ | 0, 5, 10, 30, 50 ppb |
| $CH_4$ | 1, 1.5, 2, 2.5 ppm |
| $CO_2$ | 400, 500, 700 ppm |
| RH | 5-85 % |
| T | 20-40 °C |

*Denotes that each concentration is repeated





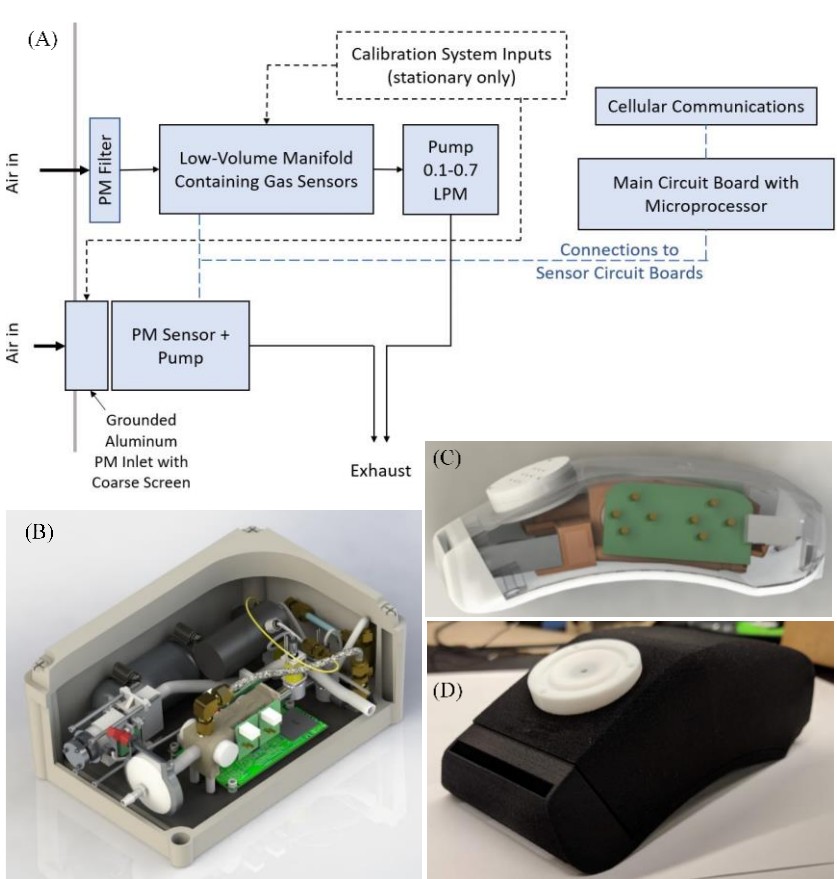


**Figure 1: Monitor designs shown as (a) simplified flow and electronics diagram, which is used in the (b) stationary (28 x 18 x 14 cm) and (c-d) portable (15 x 6.5 x 5 cm) versions of the multipollutant monitoring device. Panels b-c are solidworks renderings, and d is a photo. See Fig. S3 for additional photos.**

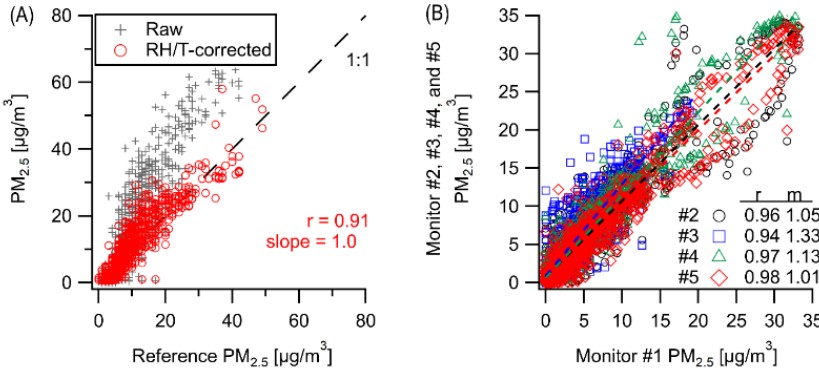

**Figure 2: Comparison of PM$_{2.5}$ concentrations between (a) our monitor and the Baltimore Oldtown site reference measurements and (b) an intercomparison of 5 co-located 5 PM sensors over 2.5 weeks in New Haven where there is a high degree of correlation with measurements even at 10-min resolution (time series data can be found in Fig. S8).**





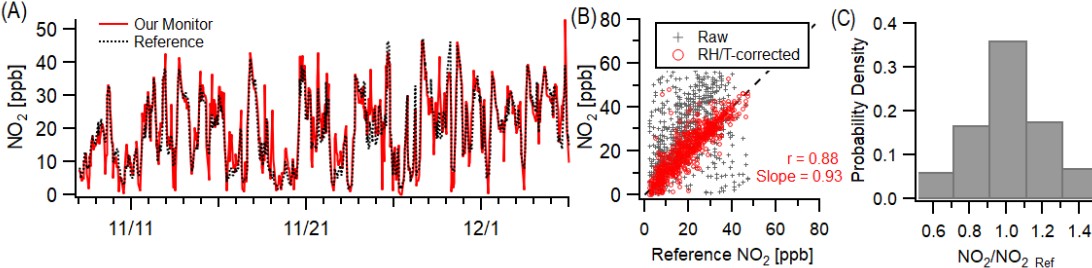

**Figure 3: Outdoor ambient monitor comparison of NO₂ from Baltimore, MD (Oldtown site). (a–b) With RH/T correction factors the data achieves good correlation. (c) Over 35% of the measurements are within 10% of the reference site.**

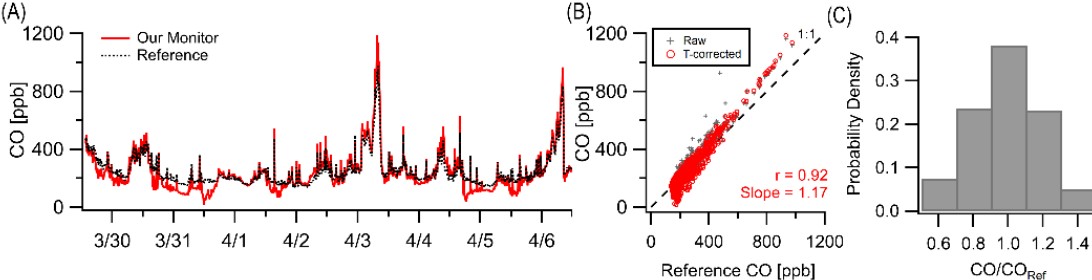

**Figure 4: (a) Outdoor ambient monitor comparison of CO from New Haven, CT. (b) Minimal overall effects from RH/T were observed (although the effect of T could be amplified at higher ambient levels, see Fig. S9). (c) At a 10-min resolution 38% of data points within 10% of the reference and 85% within 30%.**

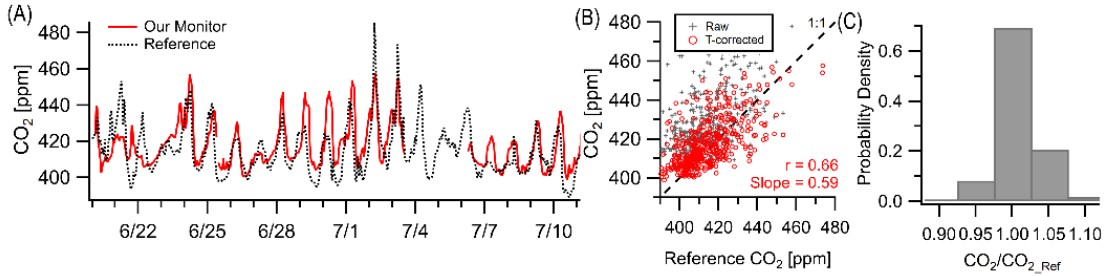

**Figure 5: (a–b) Outdoor ambient monitor comparison of CO₂ from Baltimore, MD. The monitor was located at the OWLETS campaign while the reference data came from the NIST NEB site (2.7 km away). Occasional lags in pollution episodes, potentially due to the displacement of the monitor from the reference, are seen (e.g. 6/30, 7/1, 7/7) leading to a lower correlation coefficient than other pollutants. (c) Despite this, 70% of data points fall within 2.5% of the reference.**





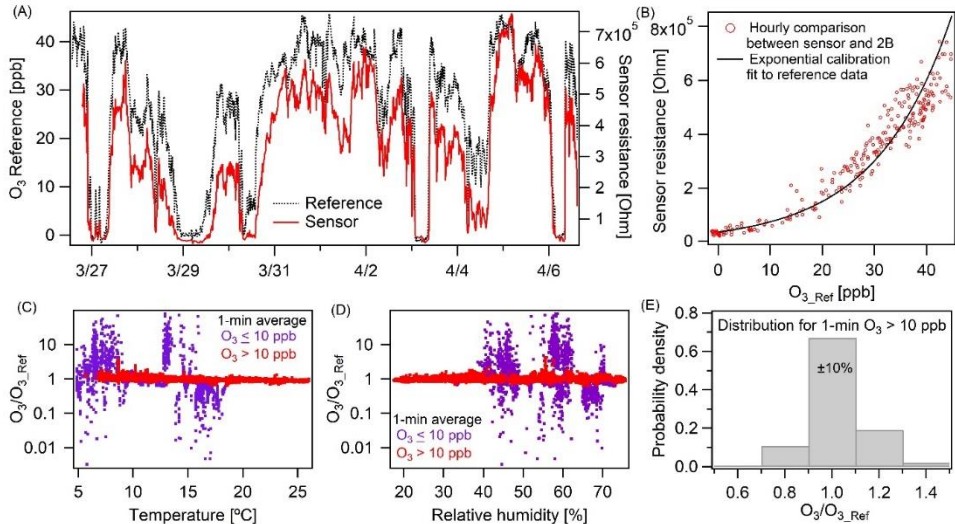


**Figure 6: Ozone calibration performance and evaluation (a) over 2 weeks in New Haven, CT shown and (b) calibrated against a 2-B Tech reference monitor (note panel A shows sensor resistance, not concentration). (c-d) The ratio of our calibrated vs. reference measurement for concentrations greater and less than 10 ppb over the range of RH and temperatures observed, with no dependence on RH and a slight temperature dependence. (c–d) At concentrations greater than 10 ppb our measurements are much more**
**accurate, (E) with 70% of the 1 min average data falling within ±10%.**

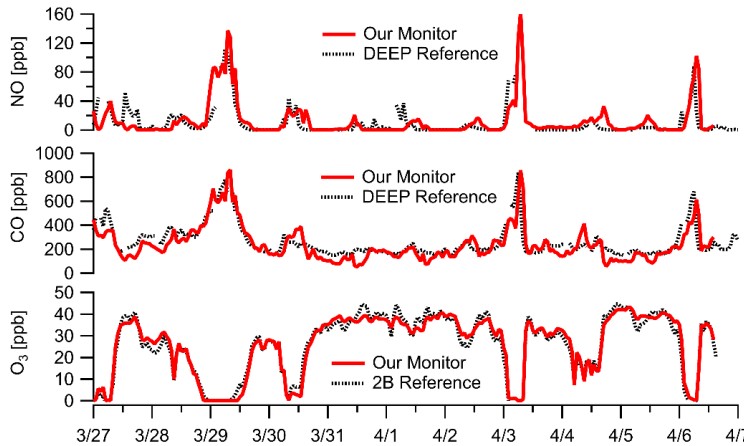

**Figure 7: Ambient data at 1 h resolution in New Haven, CT near construction shows large NO enhancements. The presence of NO is confirmed by both the titration of O₃ (i.e. NO+O₃ reaction) to zero and large enhancements in CO (a combustion co-pollutant). The near-road DEEP Criscuolo Park site (1.6 km away from sampling location in downtown New Haven) is used for comparison.**
**Note: our NO sensor does not have significant cross-response to O₃ or CO. (See Fig. S11 for additional NO comparisons.)**





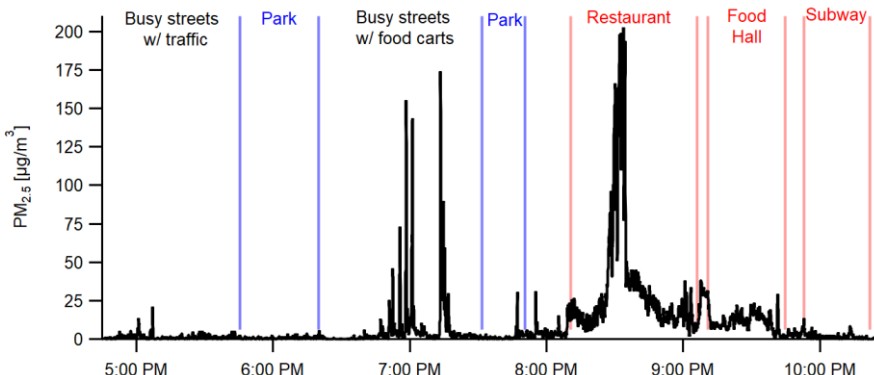

**Figure 8: Portable personal monitor data for PM$_{2.5}$ from New York City with labeled events, locations, and nearby sources. Data is shown at 1–sec resolution (with RH/T correction factors) capturing rapidly changing microenvironments such as emissions from individual food carts (6:45–7:15 PM).**

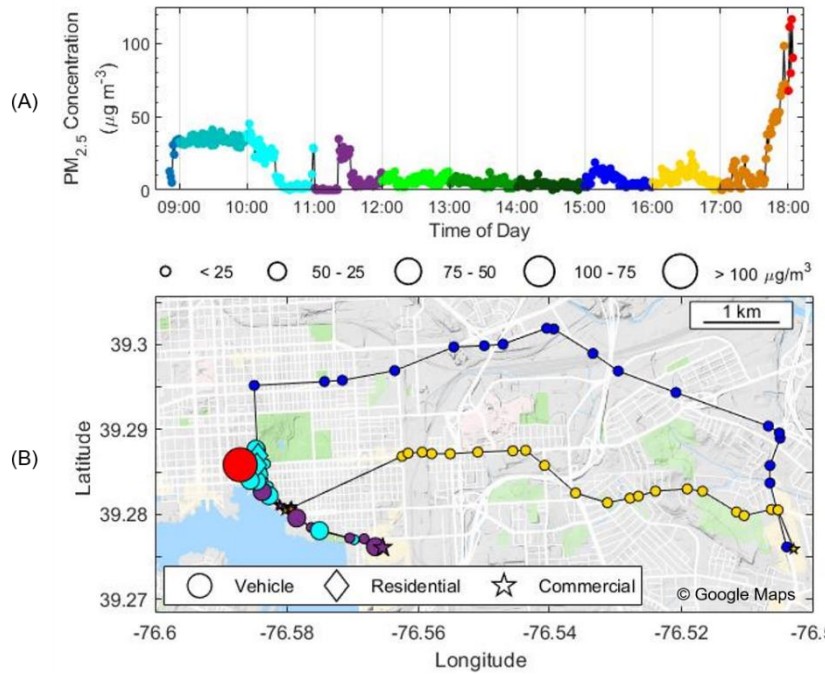


**Figure 9: PM$_{2.5}$ mass concentration (µg/m$^3$) as a function of (a) time and (b) mapped with GPS coordinates. The color of the dots in both panels change each hour to represent the time in Panel B. The background colors in Panel A and shapes in Panel B indicates microenvironment, i.e., residential (diamond), commercial (star), and vehicular (circles). The size of the points in (B) corresponds to the mass concentration.**




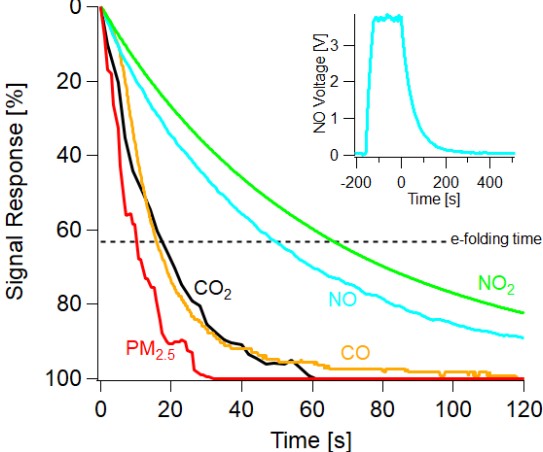

**Figure 10: Instrument response times shown as the normalized signal response of various sensors in the stationary monitor to a step change in target pollutant concentration and their respective e-folding times. Insert shows a typical step change of calibration gas and the sensor response. Time zero indicates when the multipollutant monitor switched to sampling zero air after reaching a steady state response to the pollutant. Gas sensor response times vary due to the individual sensors' diffusive or electrochemical timescales.**



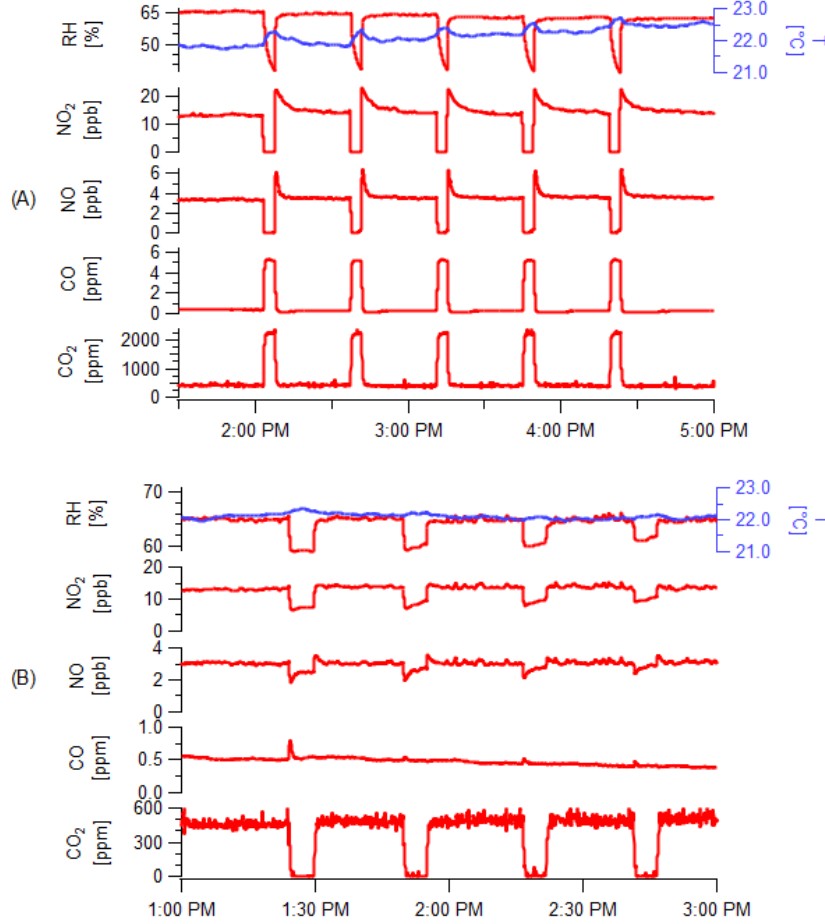

**Figure 11: Performance of the on-board calibration system over sequential cycles (for demonstration) showing (a) the span gas delivery for CO and CO₂ (and zero air to provide zero concentrations for NO and NO₂) over 5 repeated cycles and (b) the zeroing function for CO₂ over 4 cycles where other signals are shown to illustrate minimal changes with zero trap (NOₓ changes are due to shown RH changes). PM zeroing function results are shown in Fig. S13.**

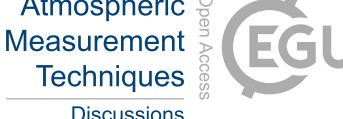
Table 3. Summary of statistics and comparison of the SEARCH stationary multipollutant monitor to several recent literature field studies.

| Pollutant | Sensor Type | Temp. Res. | Unit | m | r | r² | MBE | MAE | RMSE | Reference |
|---|---|---|---|---|---|---|---|---|---|---|
| NO₂ | Alphasense NO2-B4 | 10 min | ppb | | | 0.80 | | 5.8 | 8.3 | Bigi et al., 2018[d] |
| | Alphasense NO2-B43F | 15 min | ppb | 0.64 | | 0.67 | -0.4 | 3.48 | | Zimmerman et al., 2018 |
| | Alphasense NO2-B43F | 5 min | ppb | 0.81 | | 0.69 | 1.2 | 3.45 | 4.56 | Cross et al., 2017 |
| | **Alphasense NO2-A43F** | **1hr** | **ppb** | **0.93** | **0.88** | **0.77** | **0.8** | **3.7** | **5.3** | **This work** |
| | Alphasense NO2-B4 | 15 min | ppb | 0.38[a] | 0.49 | | 13.3 | 26.23 | 30.27 | Castell et al., 2017 |
| NO | Alphasense NO-B4 | 5 min | ppb | 0.94 | | 0.94 | 0.97 | 2.83 | 4.52 | Cross et al., 2017 |
| | Alphasense NO-B4 | 10 min | ppb | | | 0.94 | | 2.26 | 3.54 | Bigi et al., 2018[d] |
| | **Alphasense NO-A4** | **1 hr** | **ppb** | **0.86** | **0.74** | **0.54** | **1.6** | **8.5** | **16** | **This work** |
| | Alphasense NO-B4 | 15 min | ppb | 0.93[a] | | 0.86 | -0.54 | 12.48 | 16.35 | Castell et al., 2017 |
| CO | **Alphasense CO-A4** | **10 min** | **ppb** | **1.2** | **0.92** | **0.84** | **5** | **41** | **59** | **This work** |
| | Alphasense CO-B4 | 15 min | ppb | 0.88[a] | 0.6 | | -147.21 | 149.35 | 170.99 | Castell et al., 2017 |
| | Alphasense CO-B4 | 5 min | ppb | 0.94 | | 0.88 | -10.4 | 24.8 | 32.9 | Cross et al., 2017 |
| | Alphasense CO-B4 | 15 min | ppb | 0.86 | | 0.91 | 0.1 | 38 | | Zimmerman et al., 2018 |
| CO₂ | Gascard NG & S-100H | 1 hr | ppm | 0.48-0.67 | | 0.51-0.79 | | | | Spinelle et al., 2017[b] |
| | **Alphasense IRC-A1** | **1 hr** | **ppm** | **0.59** | **0.66** | **0.44** | **3.4** | **8.7** | **11** | **This work** |
| O₃ | Alphasense Ox-B431 | 15 min | ppb | 0.82 | | 0.86 | -0.14 | 3.36 | | Zimmerman et al., 2018 |
| | Alphasense Ox-B421 | 5 min | ppb | 0.47 | | 0.39 | 0.78 | 7.34 | 9.71 | Cross et al., 2017 |
| | Alphasense Ox-B421 | 15 min | ppb | 0.26[a] | 0.54 | | 6.76 | 19.87 | 22.2 | Castell et al., 2017 |
| | Alphasense Ox-B431 | 1 hr | ppb | 0.91 | | 0.89 | | | | Ripoll et al., 2019 |
| | MiCS-2614 | 1 hr | ppb | 0.87 | | 0.88 | | | | Ripoll et al., 2019 |
| | **MiCS-2614** | **1 min** | **ppb** | **0.99** | **0.97** | **0.94** | **-0.2** | **2.7** | **3.3** | **This work** |
| | **MiCS-2614** | **1 hr** | **ppb** | **1.01** | **0.98** | **0.96** | **-0.2** | **2.4** | **2.9** | **This work** |
| PM₂.₅ | TSI AirAssure | 1 hr | µg/m³ | 1.06 | | 0.73 | 3.9 | 5.6 | 7.8 | Feenstra et al., 2019[c] |
| | SainSmart P3 | 1 hr | µg/m³ | 1.48 | | 0.76 | 3.9 | 5.4 | 7.8 | Feenstra et al., 2019[c] |
| | Aeroqual AQY | 1 hr | µg/m³ | 0.98 | | 0.79 | -3.1 | 4.6 | 6.1 | Feenstra et al., 2019[c] |
| | Shinyei PM Eval. Kit | 1 hr | µg/m³ | 1.11 | | 0.74 | 0.2 | 4.4 | 6.4 | Feenstra et al., 2019[c] |
| | PurpleAir PA-II | 1 hr | µg/m³ | 1.63 | | 0.95 | 4.8 | 6.8 | 10.1 | Feenstra et al., 2019[c] |
| | AQMesh | 15 min | µg/m³ | | 0.51 | | -0.03 | 3.08 | 5.57 | Castell et al., 2017 |
| | **Plantower A003** | **1 hr** | **µg/m³** | **1.0** | **0.91** | **0.82** | **0.9** | **3.1** | **4.3** | **This work** |

[a] Slope and intercept value from unit 688150 in Table 2. [b] From correlation using an Artificial Neural Network calibration. [c] Top five performing (by r²) monitors. [d] Taken from Fig. 11 from SU009.