# Peer review of "Stationary and Portable Multipollutant Monitors for High Spatiotemporal Resolution Air Quality Studies including Online Calibration"

_Atmospheric Measurement Techniques, 2020_

## Referee Comment (RC1) · Anonymous Referee #1 · 3 Sep 2020

"Stationary and Portable Multipollutant Monitors for High Spatiotemporal Resolution Air Quality Studies including Online Calibration"

General Comments

The manuscript presents low-cost air quality sensor unit for multipollutant, including toxic gases, particulate matter and greenhouse gases (carbon dioxide and methane). The authors report detailed characterisation of two variants of the device: stationary and portable version and present some preliminary results from few field studies. In addition to the characterising the devices, the authors present an online calibration method that rely on the use of traceable gas which is incorporated into the device

with auto-calibration functionality built into the operational software of device. This manuscript is well written and presents a new approach physical calibration approach that is more comprehensive than any other work in this field of low-cost air quality sensor. The manuscript also present recommendations on best practices related to design, characterising and deployment of this type of device.

Specific comments

The authors did not address the issue of safety for the online calibration system which relies on the use of pressurised calibration cylinder (1500 psig). Under normal ambient application, the static variant of the multipollutant device may be exposed to high temperatures especially in the summer. This could pose potential safety concern for the structural integrity of the device. The authors have reported using B431 Alphasense sensor for the OX (P. 3, line 93) but the Table 1 and the text in line 90 page 3 suggest A variant was used. In addition, the authors stated that the static variant uses MiCS-2614 for O3, does this mean the static and portable unit has both OX and O3 measurements? The authors need to clarify this ambiguity. I recommend the authors annotate figure 1(b) and figure S3 (a & b) with labels showing main components of the photo presented. Technical corrections (main manuscript) P. 10, line 297, the phrase "...by the Plantower sensor" sound like the authors are referring to the reference device rather than the MPM device Suggest something like ".......by the multipollutant device collocated with the reference at the Baltimore Oldtown" P. 17, line 491: there is a red font in the text. P. 17, line 494: remove the "of" in the sentence. P. 23, line 625 (Figure 3 (a)) add the RH/T corrected to the legend of the time series. Ditto for figures 4(a) and 5(a) P. 25, figure 8 and 9 captions should include the dates for this deployment.

Technical corrections (supplementary information)

P. 4, line 85, the phrase "... Eqn S2 and Eqn S3" should read "... Eqn S3 and Eqn S4" P. 9, Figure S9 caption should include the temperature range for the two plots (< 18 and > 18 degree C).

---

## Referee Comment (RC2) · Anonymous Referee #2 · 13 Oct 2020

This manuscript presents the design of a new low-cost sensor system that incorporates an innovative approach to maintaining sensor performance – an automated system for performing zero/span calibrations. This manuscript is particularly relevant as the integration of a zero/span system into a low-cost sensor device has the potential to help address the ongoing challenges of sensor drift and degradation. The authors provide a detailed description of the technology, including the sensors incorporated as well as the hardware and software. Two versions of the device have been created to accommodate stationary and portable monitoring. The authors also share initial results on performance of the device and compare these results to previous studies.

[Figure]

Comments:

1. Multiple studies have observed that calibrating sensors using field co-location data as opposed to a laboratory approach tends to result in higher performing and more robust calibrations (e.g., the study mentioned in lines 56-57). Considering this information, it is recommended that the authors discuss their rationale for using a laboratory calibration approach (lines 200 -201). In addition, do the authors anticipate any limitations regarding the zero/span system given that the span gas will not have the same mixture of background pollutants as the ambient air being sampled?

2. On line 67 the authors mention the issue of sensor drift – given the wide range of sensor types used, it is recommended that the authors expand this discussion to include how drift and cross-sensitivities tend to vary by sensor type.

3. Regarding the modification to the Figaro 2600, according to the results in Figure S6 – even though the trends are linear, it appears that adding the charcoal filter seems to reduce the sensor's sensitivity, are there any other impacts from this modification, such as a slowed response time? Recommend adding more discussion on this topic.

4. Were the portable monitors also tested via a co-location in the field with reference instruments, if not could the authors discuss the rationale?

5. In Section 3.4, the authors discuss the performance of the zero/span system. It is recommended that the authors comment on the impact of the system on sensor performance for each pollutant. During the field co-location tests, did the implementation of the system have any clear impact on the resulting data? It seems this system might be most useful during long-term deployments - though the field tests appear to range between 1 week to 1 month long.

6. In Section 3.5, for Table 3, it is recommended that the authors add columns or a second table that lists the length of the deployment, the location of the deployment, and the calibration approach used (i.e., field or lab). In addition, consider highlighting

or adding comparisons to results for which a laboratory calibration was tested in the field.

7. In Section 3.5, in Table 3, for all pollutants except PM2.5 the raw OEM sensors are listed, whereas the names of commercial devices are listed for PM2.5 – recommend adding the raw OEM sensor used for all devices.

8. Reconsider the use of the term 'monitor' throughout the manuscript (i.e., multipollutant monitor and portable monitor). These devices are typically referred to as a sensor, sensor platform, or sensor system, while generally the term monitor is reserved for research or regulatory grade instruments.

9. There is some red text on line 491.

---

## Author Comment (AC1) · 17 Nov 2020

**Referee #1:**

The manuscript presents low-cost air quality sensor unit for multipollutant, including toxic gases, particulate matter and greenhouse gases (carbon dioxide and methane). The authors report detailed characterisation of two variants of the device: stationary and portable version and present some preliminary results from few field studies. In addition to the characterising the devices, the authors present an online calibration method that rely on the use of traceable gas which is incorporated into the device with auto-calibration functionality built into the operational software of device. This manuscript is well written and presents a new approach physical calibration approach that is more comprehensive than any other work in this field of low-cost air quality sensor. The manuscript also present recommendations on best practices related to design, characterising and deployment of this type of device.

- The authors did not address the issue of safety for the online calibration system which relies on the use of pressurised calibration cylinder (1500 psig). Under normal ambient application, the static variant of the multipollutant device may be exposed to high temperatures especially in the summer. This could pose potential safety concern for the structural integrity of the device.

Response: Thank you for identifying this missing element in the text. It was however something we considered extensively in the design of the device. The cylinder, valves, and associated fittings are selected based on their specifications to withstand the cylinder pressure (reference: Swagelok/TESCOM specifications). For example, the pressurized gas calibration cylinder is constructed with 304L stainless steel with a DOT-3E1800 specification. This specification allows for a maximum working pressure of 1800 psig between -53°C to 37°C, which is higher than our use of 1500 psig. Elevated summertime temperatures are indeed one consideration. The specification calls for a maximum working pressure of 1360 psig at a temperature of 93°C, but based on summertime temperatures in our testing, the monitors do not go over 60°C, and best monitor siting practices suggests shading of the monitor to further mitigate elevated temperatures (as discussed in Section 3.6).

We have added text to the manuscript (lines 251-256) to address this point, and instruct future researchers to pay attention to component specifications and the standard gas concentrations in the cylinder. Furthermore, labels within the device are also used to warn users about the presence of pressurized gas.

- The authors have reported using B431 Alphasense sensor for the OX (P. 3, line 93) but the Table 1 and the text in line 90 page 3 suggest a variant was used. In addition, the authors stated that the static variant uses MiCS-2614 for O3, does this mean the static and portable unit has both OX and O3 measurements? The authors need to clarify this ambiguity.

Response: In the manuscript, the stationary version of the monitor uses the MiCS-2614 sensor to measure  $O_3$  while the portable monitor uses the Alphasense A431 sensor to measure  $O_3$ . (via the  $NO_2$  subtraction specified by Alphasense). However, the 3D printed gas manifold was designed to be able to accommodate the Alphasense A431 sensor in the stationary monitor if necessary given the uniform size of the Alphasense A-series sensors. For this study, the Alphasense A431

sensor was solely used in the portable monitor. To reduce any confusion or ambiguity, we have removed the  $O_x$  listing on line 93 and added a column to Table 1 to specify which sensors are used in each version.

- I recommend the authors annotate figure 1(b) and figure S3 (a & b) with labels showing main components of the photo presented.

Response: Thank you for the suggestion. We have added annotations for the main components (e.g. gas inlet, online calibration cylinder) in Figure 1 and Figure S3.

- P. 10, line 297, the phrase ". . .by the Plantower sensor" sound like the authors are referring to the reference device rather than the MPM device Suggest something like ". . . . .by the multipollutant device collocated with the reference at the Baltimore Oldtown"

Response: Thank you for the comment, we have changed the wording to be less ambiguous.

- P. 17, line 491: there is a red font in the text.

Response: Fixed, thank you.

- P. 17, line 494: remove the "of" in the sentence.

Response: Fixed, thank you.

- P. 23, line 625 (Figure 3 (a)) add the RH/T corrected to the legend of the time series. Ditto for figures 4(a) and 5(a)

Response: We have added language to the caption to indicate that "Our Monitor" in the (a) panel time series for Figures 3, 4, and 5 is the corrected, not raw, data.

- P. 25, figure 8 and 9 captions should include the dates for this deployment.

Response: We have added the deployment dates in the captions of figure 8 and 9. The NYC deployment occurred on June  $23^{rd}$ , 2018 and the Baltimore deployment occurred on March  $2^{nd}$ , 2019.

- P. 4, line 85, the phrase "... Eqn S2 and Eqn S3" should read "... Eqn S3 and Eqn S4"

Response: Fixed, thank you.

- P. 9, Figure S9 caption should include the temperature range for the two plots (< 18 and > 18 degree C). *Response: Fixed, thank you.*

---

## Author Comment (AC2) · 17 Nov 2020

**Referee #2:**

This manuscript presents the design of a new low-cost sensor system that incorporates an innovative approach to maintaining sensor performance – an automated system for performing zero/span calibrations. This manuscript is particularly relevant as the integration of a zero/span system into a low-cost sensor device has the potential to help address the ongoing challenges of sensor drift and degradation. The authors provide a detailed description of the technology, including the sensors incorporated as well as the hardware and software. Two versions of the device have been created to accommodate stationary and portable monitoring. The authors also share initial results on performance of the device and compare these results to previous studies.

- Multiple studies have observed that calibrating sensors using field co-location data as opposed to a laboratory approach tends to result in higher performing and more robust calibrations (e.g., the study mentioned in lines 56-57). Considering this information, it is recommended that the authors discuss their rationale for using a laboratory calibration approach (lines 200 -201). In addition, do the authors anticipate any limitations regarding the zero/span system given that the span gas will not have the same mixture of background pollutants as the ambient air being sampled?

*Response: We agree with the reviewer that field co-location is a very effective approach to sensor calibration. In fact, in the larger project that uses the monitors described in this study, we employ a hybrid approach to calibration that utilizes co-location as part of the strategy. As noted in Section S1, we use laboratory calibrations to get initial sensor calibration and quality control performance, as well as develop field environmental correction factors with each type of sensor for relative humidity and temperature (where appropriate) that build off of the laboratory calibrations. For example, in a recent paper we employ co-location approaches to develop a statistical calibration method for the $PM_{2.5}$ sensor used in our multipollutant monitor (Datta et al., 2020; Atoms. Env., doi: 10.1016/j.atmosenv.2020.117761). The on-board system is designed for future longer deployments to maintain better calibrations and track sensor response/zero drift during the in-field period after lab calibration and/or co-location with reference monitors. A few limitations of the zero/span system described in the paper are the number of reactive gases one can effectively combine into a single standard mixture and the available materials for room temperature removal of pollutants in the zero trap. Another limitation is the potential for insufficient cylinder pressure as the tank approaches empty, which can be tracked by instillation date and calibration frequency, as well as monitoring calibration data. At current rates the cylinder can operate for over a year before needing to be refilled. Other limitations are addressed by maintaining acceptable relative humidity during the zero and span functions (e.g. water permeation setup) and using variable calibration schedules to get variations in calibration temperature. Additionally, the online data visualization platform we utilize (Grafana) also allows us to remotely monitor calibrations and replace poorly performing monitors in near real time. Without this, we would need to wait until the deployment ended to check for monitor performance. We acknowledge that the background mixture is going to be different during the span checks (less so during the zero check), but that is not unlike calibration with typical instruments/standards, and has the advantage of reducing confounding factors that could*

*introduce additional variables that would affect the ability to track drift over time. To address the reviewer's comment, we have examined and revised relevant sections of the text to ensure this discussion is accurately reflected (Section 2.1.4).*

- On line 67 the authors mention the issue of sensor drift – given the wide range of sensor types used, it is recommended that the authors expand this discussion to include how drift and cross-sensitivities tend to vary by sensor type.

*Response: We agree that understanding sensor drift and cross-sensitivities to other pollutants are key for low cost sensor deployment. To address the reviewer's request, we have added a couple sentence summary in the sensor methods section and referenced a more comprehensive review of low-cost air quality sensors from the World Meteorological Organization which includes discussion of sensor drift and common cross-sensitivities (in the introduction at line 66-68 and in the Section 2.1.1).*

- Regarding the modification to the Figaro 2600, according to the results in Figure S6 – even though the trends are linear, it appears that adding the charcoal filter seems to reduce the sensor's sensitivity, are there any other impacts from this modification, such as a slowed response time? Recommend adding more discussion on this topic.

*Response: As noted in the reviewer's comment, the use of the activated carbon filter slightly reduced the total sensor response (while remaining linear) to $CH_4$ concentrations tested in the 1.4-2.4 ppm range, which may be attributed to decreases in background VOCs. It is possible that there may be a very slight increase in sensor response time due to the added diffusion length for gases, but that was not observed here and is not expected to be an issue with minute resolution data averages. To ensure that accurate response factors are generated in conditions similar to ambient measurements, all calibrations are performed with the activated carbon filter in place. In Section 2.1.3 we discuss these and other potential issues arising from the use of using the charcoal filter. For example, we implement a 3D-printed PLA shell to keep the filter in place which could react with pollutants of interest (e.g. ozone). Teflon "tape" is placed around the filter in order to reduce interactions between the shell and reactive gases as well as prevent any charcoal fibers from shedding the inside the gas manifold.*

- Were the portable monitors also tested via a co-location in the field with reference instruments, if not could the authors discuss the rationale?

*Response: For this study, the portable monitors were not co-located in the field since they largely consist of the same sensors (exception: Alphasense Ox sensor) and a similar physical design that was scaled down. We focused evaluation efforts on the stationary multipollutant monitor that had a larger set of sensors, which are included as part of the portable monitor. It was not viewed as necessary to co-locate the portable configuration as well, especially considering it is not constructed with the same extent of weatherproofing. In the manuscript we focus on $PM_{2.5}$ as a proof of concept for high spatiotemporal personal exposure studies which uses the same Plantower sensor used in the stationary model.*

- In Section 3.4, the authors discuss the performance of the zero/span system. It is recommended that the authors comment on the impact of the system on sensor performance for each pollutant. During the field co-location tests, did the implementation of the system have any clear impact on the resulting data? It seems this system might be most useful during long-term deployments - though the field tests appear to range between 1 week to 1 month long.

*Response: We agree with reviewer that this is an interesting area for ongoing work and evaluation of the system and that it will be most useful for longer-term deployments (e.g. months to 1+ year) where drift with sensor aging will be more pronounced. Unfortunately, given the duration of the measurements used in this manuscript, we are unable to include this additional analysis/discussion, but we have added that future work in the field should examine the effect of on-board calibration systems on long-term sensor performance. We have updated language in Section 2.1.4 to address this concern.*

- In Section 3.5, for Table 3, it is recommended that the authors add columns or a second table that lists the length of the deployment, the location of the deployment, and the calibration approach used (i.e., field or lab). In addition, consider highlighting C2 or adding comparisons to results for which a laboratory calibration was tested in the field.

*Response: We agree that the length of deployment and location of deployment is of great importance when comparing between low cost sensors. We have broken up Table 3 into two tables now to allocate additional space for deployment length and deployment location. Additionally, two additional studies have been added to the comparison (the Berkeley Atmospheric $CO_2$ Observation Network). Language related to this addition has been updated in Section 3.5 (Comparison with Literature).*

- In Section 3.5, in Table 3, for all pollutants except PM2.5 the raw OEM sensors are listed, whereas the names of commercial devices are listed for PM2.5 – recommend adding the raw OEM sensor used for all devices.

*Response: Thank you for the suggestion, we have updated to include OEM sensor data where appropriate in Table 4.*

- Reconsider the use of the term 'monitor' throughout the manuscript (i.e., multipollutant monitor and portable monitor). These devices are typically referred to as a sensor, sensor platform, or sensor system, while generally the term monitor is reserved for research or regulatory grade instruments.

*Response: We appreciate the reviewer's attention to word choice. To avoid confusion, we deliberately used the term "monitor" in the manuscript to represent the collection of various sensors together in one "multipollutant monitor" along with all of the other components (e.g. flow and calibration systems) and use the term "sensor" to refer to the individual sensing components (e.g. electrochemical sensor, optical sensor) which have their own dedicated sections. This was done to represent the device as a collection of a wide range of different sensor types and suppliers, and not to suggest that there is a single sensing component measuring all the pollutants listed. Based on a survey of other papers, popular devices and the language used*

*to describe them on the AQMD AQ-Spec and EPA Air Sensor Toolbox, the terms monitor and sensor are used somewhat interchangeably to describe the overall devices (e.g. ELM monitor, Aeroqual monitor) and the term "reference monitor" is used elsewhere to describe the regulatory instruments they are compared to. For example, in Atmospheric Measurement Techniques there have been several recent articles which use the term "monitor" in the same manner as in this paper. In* Malings et al. (2019) *the researchers developed a "Real-time Affordable Multi-Pollutant (RAMP) monitor" comprised of low cost sensors similar to those in our multipollutant monitor. Another example comes from* Wendt et al. (2019)*, who design a low cost monitor for measuring $PM_{2.5}$ and aerosol optical depth (AOD). Yet, we do acknowledge these the term "sensor" is typically used to describe low-cost non-regulatory methods, and we do not intend to misrepresent the devices as federal reference methods. To address the reviewer's comment we have added the following language in the introduction to clarify the word choice at the outset: "Here the term "monitor" or "multipollutant monitor" is used to describe the collection of sensors and other components (e.g. flow channels, valves, online calibration system) used while the term "sensor" is used to describe the standalone sensing components." (Section 2.1.1, lines 76-78).*

*References:*
Malings, C., Tanzer, R., Hauryliuk, A., Kumar, S. P. N., Zimmerman, N., Kara, L. B., Presto, A. A. and Subramanian, R.: Development of a general calibration model and long-term performance evaluation of low-cost sensors for air pollutant gas monitoring, Atmos. Meas. Tech., 12(2), 903–920, doi:10.5194/amt-12-903-2019, 2019.
Wendt, E. A., Quinn, C. W., Miller-Lionberg, D. D., Tryner, J., L'Orange, C., Ford, B., Yalin, A. P., Jeffrey, P., Jathar, S. and Volckens, J.: A low-cost monitor for simultaneous measurement of fine particulate matter and aerosol optical depth - Part 1: Specifications and testing, Atmos. Meas. Tech., 12(10), 5431–5441, doi:10.5194/amt-12-5431-2019, 2019.

- There is some red text on line 491.

*Response: Fixed, thank you.*